# TIGHT LOWER BOUNDS FOR DIFFERENTIALLY PRIVATE ERM

## ABSTRACT

We consider the lower bounds of differentially private ERM for general convex functions. For approximate-DP, the well-known upper bound of DP-ERM is $O(\frac{\sqrt{p\log(1/\delta)}}{\epsilon n})$, which is believed to be tight. However, current lower bounds are off by some logarithmic terms, in particular $\Omega(\frac{\sqrt{p}}{\epsilon n})$ for constrained case and $\Omega(\frac{\sqrt{p}}{\epsilon n \log p})$ for unconstrained case. We achieve tight $\Omega(\frac{\sqrt{p\log(1/\delta)}}{\epsilon n})$ lower bounds for both cases by introducing a novel biased mean property for fingerprinting codes.

As for pure-DP, we utilize a novel $\ell_2$ loss function instead of linear functions considered by previous papers, and achieve the first (tight) $\Omega(\frac{p}{\epsilon n})$ lower bound. We also introduce an auxiliary dimension to simplify the computation brought by $\ell_2$ loss.

Our results close a gap in our understanding of DP-ERM by presenting the fundamental limits. Our techniques may be of independent interest, which help enrich the tools so that it readily applies to problems that are not (easily) reducible from one-way marginals.

## 1 INTRODUCTION

Since the seminal work of Dwork et al. (2006), differential privacy (DP) has become the standard and rigorous notion of privacy guarantee for machine learning algorithms, among which many fundamental ones are based on empirical risk minimization (ERM). Motivated by this, private ERM becomes one of the most well-studied problem in the DP literature, e.g. Chaudhuri and Monteleoni (2008); Rubinstein et al. (2009); Chaudhuri et al. (2011); Kifer et al. (2012); Song et al. (2013); Jain and Thakurta (2014); Bassily et al. (2014); Talwar et al. (2015); Kasiviswanathan and Jin (2016); Fukuchi et al. (2017); Wu et al. (2017); Zhang et al. (2017); Wang et al. (2017); Iyengar et al. (2019); Bassily et al. (2020); Kulkarni et al. (2021); Asi et al. (2021); Bassily et al. (2021); Wang et al. (2021).

Roughly speaking, in the ERM setting, we are given a convex function family defined on a convex set $\mathcal{C} \subseteq \mathbb{R}^p$ and a sample set $\mathcal{D} = \{d_1, \cdots, d_n\}$ drawn i.i.d from some unknown distribution $\mathcal{P}$ with the objective to minimize the loss function

$$L(\theta; \mathcal{D}) = \frac{1}{n} \sum_{i=1}^{n} \ell(\theta; d_i),$$

and the value $L(\theta; \mathcal{D}) - \min_{\theta' \in \mathcal{C}} L(\theta'; \mathcal{D})$ is called the excess empirical loss with respect to solution $\theta$, measuring how it compares with the best solution in $\mathcal{C}$.

Private ERM in the constrained case was studied first and most of the previous literature belongs to this case. More specifically, the constrained case considers convex loss functions defined on a bounded convex set $\mathcal{C} \subsetneq \mathbb{R}^p$. Assuming the functions are 1-Lipschitz over the convex set of diameter 1, the $\Omega(\frac{\sqrt{p}}{\epsilon n})$ lower bound of private ERM is given by Bassily et al. (2014), even for (special and simpler) generalized linear model (GLM).

However, there are still several aspects that existing works don't cover. First, existing upper bounds are off by at least a logarithmic term $\sqrt{\log(1/\delta)}$. For example in Wang et al. (2017); Bassily et al.

(2019) they give upper bounds like $O(\frac{L||\theta_0 - \theta^*||_2 \sqrt{p \log(1/\delta)}}{\epsilon n})$, which are believed to be tight. In Bassily et al. (2014), they present a lower bound $\Omega(\frac{\sqrt{p}}{n\epsilon})$ by reducing linear loss to one-way marginal results in Hardt and Talwar (2010); Bun et al. (2018). In Steinke and Ullman (2015) they achieve the tight lower bound for answering one-way marginals with respect to $\ell_1$ norm, but it does not imply tight lower bounds for general loss functions by the methods in Bassily et al. (2014) directly.

Another aspect is DP-ERM in the unconstrained case which was neglected before and gathered people's attention recently, see Jain and Thakurta (2014); Song et al. (2021). The unconstrained case is interesting in that we can't use linear loss any more which lies at the heart of the construction in the constrained case. Moreover, previous algorithms in the constrained case suffer from the curse of dimensionality when $p$ is large. For example, when $p = \Omega(n^2)$, the lower bound is $\Omega(1)$ and any private algorithm can not get meaningful bounds on the excess empirical loss. Song et al. (2021) proves an dimension-independent $O(\frac{\sqrt{\text{rank}}}{\epsilon n})$ upper bound in the unconstrained setting for the special case of GLMs, where rank denotes the rank of the feature matrix of GLM. However, it's unknown if similar results can be achieved for general loss functions. As for the lower bound, Asi et al. (2021) give $\Omega(\frac{\sqrt{p}}{n\epsilon \log p})$ lower bound by considering $\ell_1$ loss as the objective functions and reducing the results also from one-way marginals.

## 1.1 OUR CONTRIBUTIONS

In this paper, we fill up the two gaps together by proving an $\Omega(\min(1, \frac{\sqrt{p \log(1/\delta)}}{n\epsilon}))$ tight lower bound for the excess risk of unconstrained 1-Lipschitz convex loss functions for approximate differentially private algorithms. This bound is automatically applicable in the constrained case, which improves previous results and achieve a tight lower bound for both constrained and unconstrained case. We summarize our main results as follows:

- We prove an $\Omega(\min(1, \frac{\sqrt{p \log(1/\delta)}}{n\epsilon}))$ tight lower bound for the excess risk of unconstrained 1-Lipschitz convex loss functions for approximate differentially private algorithm. This bound improves Asi et al. (2021) by a $\log(p)\sqrt{\log(1/\delta)}$ factor in the unconstrained case and matches the upper bound in Kairouz et al. (2020).
- We also prove an $\Omega(\min(1, \frac{p}{n\epsilon}))$ lower bound for the excess risk of unconstrained 1-Lipschitz convex loss functions for any pure differential privacy algorithm.

Note that our main results for unconstrained case can be extended to constrained case directly, thus our lower bound for approximate private algorithm is $\sqrt{\log(1/\delta)}$ multiplicative better than the well-known bound in Bassily et al. (2014) with the help of group privacy technique in Steinke and Ullman (2015).

A key contribution of this paper is novel tools for private lower bound techniques. For most problems, accuracy lower bound in the private setting is established via reduction from one-way marginals. Hence the tools for lower bounds is quite limited. We contribute to refinement of such tools – in particular, we propose modifications such as the additional "biased means" property to fingerprinting codes, which is the key lower bound technique. Such modifications help enrich the tools so that it readily applies to problems which are not (easily) reducible from one-way marginals.

## 1.2 OUR TECHNIQUES

In general, the direct technical challenge of the unconstrained case lies in the choice of loss function and the difficulties caused by the new loss function. The loss function is required to be convex and Lipschitz-continuous at the same time. In the constrained case, the linear loss function is obviously a good choice for constructing lower bounds, because it is easy to analyze and easily reducible from one-way marginals. However, in the unconstrained case any non-trivial linear loss function can take '$-\infty$' value thus not applicable anymore. We further observe that convexity plus Lipschitz-continuity means 'asymptotically linear': the sub-gradient along any direction must converge. This observation guides us in choosing new loss functions for unconstrained DP-ERM (which can be extended to constrained case directly). We briefly introduce the new problems caused by the new loss functions and our method to overcome them for approximate-DP and pure-DP separately.

### 1.2.1 APPROXIMATE-DP

The construction of our lower bound for approximate-DP is based on the Fingerprinting Codes, which was first studied by Wagner (1983) and developed by Boneh and Shaw (1998); Tardos (2008).

From a technical perspective, we change the previously used linear loss and use an $\ell_1$ norm function instead where $\ell(\theta; d) = \|\theta - d\|_1$. $\ell_1$ loss has been used in a concurrent work Asi et al. (2021) which proves an $\Omega(\frac{\sqrt{p}}{\epsilon n \log p})$ lower bound of approximate DP in the constrained case by reducing the results from one-way marginals, and can be extended to unconstrained case directly. We improve this bound by logarithmic terms and achieve optimality by utilizing the group privacy technique from Steinke and Ullman (2015). We observe a novel biased mean property in the fingerprinting code to successfully combine and adjust these techniques to fit the $\ell_1$ loss.

We briefly describe the proof based on group privacy technique first. In the group privacy we need to copy some hard instances of data-set $\mathcal{D}^k$ of size $n_k := \lfloor n/k \rfloor$ according to the construction of fingerprinting codes by $k$ times, and append $n - k n_k$ data points to get a final data-set $\mathcal{D}$ of size $n$. Fix any $(\epsilon, \delta)$-differentially private algorithm $\mathcal{A}$ for $\mathcal{D}$, if we remove one element $i^*$ from $\mathcal{D}^k$, we can get $\mathcal{D}^k_{-i^*}$ and $\mathcal{D}_{-i^*}$ where $\mathcal{D}_{-i^*}$ and $\mathcal{D}$ can have at most $k$ elements different. Running $\mathcal{A}$ on $\mathcal{D}$ and $\mathcal{D}_{-i^*}$ respectively, we get an $(k\epsilon, \delta')$-differential privacy algorithm for $\mathcal{D}^k$ and $\mathcal{D}^k_{-i^*}$. Setting $k$ appropriately, if $\mathcal{A}$ can lead to small error on the DP-ERM, it can be an adversary which contradicts the properties of the fingerprinting codes. Intuitively, the differential privacy means it is hard to find the removed element $i^*$, but fingerprinting codes suggest the removed element is traceable as long as $\mathcal{D}^k$ satisfies the required properties and $\mathcal{A}$ leads to small excess empirical loss with respect to $L(\theta; \mathcal{D}^k)$.

The direct use of the biased mean property is in appending the $n - k n_k$ points. As nearly all of previous lower bounds in DP convex optimization are based on the results from one-way marginals, we try to demonstrate the proof in the language of one-way marginals. Because for linear functions, large one-way marginal errors lead to large excess empirical loss directly, which means the lower bound of private one-way marginals can apply to DP-ERM. But it is obvious that without additional assumption, large one-way marginal errors can not mean large excess empirical loss of the $\ell_1$ loss functions anymore. Consider the toy case when $p = 1$ and $\mathcal{D}^k = \{d_i\}_{i=1}^{\lfloor n/k \rfloor}$ where $d_i \in \{0, 1\}$. Denote the mean of these $\lfloor n/k \rfloor$ by $\overline{\mathcal{D}^k}$. Similarly let $\overline{\mathcal{D}}$ be the mean values of $\mathcal{D}$ constructed from $\mathcal{D}^k$ by method above. For example, if $\overline{\mathcal{D}^k} = 1/2$ and we only append points $1/2$, then $\overline{\mathcal{D}} = 1/2$ and whatever the one-way marginals are, the excess empirical loss (of the $n$ $\ell_1$ loss functions) can be 0, as $L(\theta; \mathcal{D}) = \frac{1}{n} \sum_{i=1}^n \|\theta - d_i\|_1$ is a constant function over $[0, 1]$. So we need the mean $\overline{\mathcal{D}^k}$ to be biased. More specifically, we need $|\overline{\mathcal{D}^k} - 1/2|$ should be larger than some value depending on $k$, then we can append $n - k n_k$ dummy points safely. For general $p$, we need the unbiased mean property holds for a large fraction of coordinates. For any single dimension, the biased mean property serves to ensure the prediction of the column (dimension) is unchanged during the group privacy mechanism, in which some number of dummy points are appended that may potentially change the prediction of unbiased column. This novel property sets more stringent conditions. Fortunately, we observe that the previous construction of fingerprinting codes in Bun et al. (2018) satisfies it.

### 1.2.2 PURE-DP

Although the square loss seems tempting in the constrained case, which intuitively reduces the unconstrained case to constrained because the loss value grows fast outside a bounded region and also makes computation simple. It's unfortunately non-Lipschitz in the unconstrained case thus not applicable directly.

We use the novel $\ell_2$-norm loss as a natural substitute, which is both convex and Lipschitz-continuous. Unlike the constrained case, the $\ell_2$-norm loss brings the drawback that the minimizer of the ERM problem no longer has a closed form solution or any nice property for computation. Roughly speaking, in the analysis of Bassily et al. (2014) they have an 'adding dummy points' procedure which will perturb the minimizer. Linear loss has this nice property that after adding these dummy points the minimizer can only move along its direction, while for $\ell_2$ loss the minimizer might be intractable. To overcome this problem, we define the Fermat point and introduce an auxiliary dimension to simplify the messy calculation brought by $\ell_2$-norm loss in our analysis. The

dummy points have support only in this auxiliary dimension and guarantee that the perturbation of the minimizer is still along its own direction, reducing computation in any high dimension to a two-dimension subspace spanned by the minimizer and the auxiliary dimension.

### 1.3 CONSTRAINED AND UNCONSTRAINED

In this subsection, we briefly discuss the relationships and differences between constrained case and unconstrained case, and compare our bounds with previous bounds.

Previous studies on DP-ERM mostly focus on the constrained setting, and the unconstrained case recently attract people's interest because Jain and Thakurta (2014); Song et al. (2021) found that an $O(\frac{\sqrt{\mathrm{rank}}}{\epsilon n})$ upper bound can be achieved for minimizing the excess risk of GLMs, which evades the curse of dimensionality.

It has been known that the unconstrained condition is necessary for dimension independence, as pointed out by Bassily et al. (2014) in which they prove an $\Omega(\frac{\sqrt{p}}{n\epsilon})$ lower bound even for minimizing constrained GLMs for the case when "rank $\leq n \ll p$".

We are interested in the necessity of the unconstrained condition to get rank-dependent bound. The unconstrained GLM can be viewed as a rank-dimensional problem, as the noise added in the null space of the feature matrix will not affect the excess empirical loss. However, this does not hold in the constrained case. Take the dimension-independent algorithm in Song et al. (2021) which is based on SGD as an example. The pitfall for the dimension-independent algorithm lies in projection if SGD is modified to projected-SGD for constrained case, that running SGD in the constrained setting requires projection which might "increase rank". We can see there is some fundamental difference between constrained and unconstrained case, and analyzing unconstrained case is also an interesting and important direction.

Classic methods, like Bassily et al. (2014) usually connect linear loss to one-way marginals Bun et al. (2018), and then use lower bounds for one-way marginals to imply lower bounds for linear loss. As Bassily et al. (2014) are using results from Hardt and Talwar (2010); Bun et al. (2018) in one-way marginals and Steinke and Ullman (2015) achieves tight $\Omega(\frac{\sqrt{p \log(1/\delta)}}{\alpha\epsilon})$ bound by using the novel group privacy technique, one may ask whether combining Steinke and Ullman (2015) and Bassily et al. (2014) can achieve the tight lower bound in the constrained case trivially. Because Steinke and Ullman (2015) considers $\ell_1$ distance but Bassily et al. (2014) considers $\ell_2$ distance, there is a $\sqrt{p}$ gap between them and one can't directly combine them. Though with some effort, one may get tighter bounds in the constrained case by modifying the results in Steinke and Ullman (2015) from $\ell_1$ norm to $\ell_2$ norm, then applying the analysis in Bassily et al. (2014).

As shown in Asi et al. (2021), proving a nearly tight lower bound in the unconstrained case is direct by utilizing one-way marginals and choosing the right objective functions, but getting rid off those extra logarithmic terms in the unconstrained case is nontrivial as the one-way marginals can not work directly in group privacy. To the best of our knowledge, our result is the first time that achieves this improved tight lower bound for general loss function class in both cases. See Table 1.4 for detailed comparisons between previous bounds and ours.

### 1.4 RELATED WORK

The existing lower bounds of excess empirical loss, i.e. the constrained case in Bassily et al. (2014) and the unconstrained case in Song et al. (2021), are all using GLM functions. The objective function used in Bassily et al. (2014) is $\ell(\theta; d) = \langle \theta, d \rangle$ which can't be applied in the unconstrained case, otherwise the loss value would be infinite. Considering this limitation, Song et al. (2021) adopts $\ell(\theta; d) = |\langle \theta, x \rangle - y|$. They transfer the problem of minimizing GLM to estimating one-way marginals, and then get the lower bound by properties in the definition of the Fingerprinting Codes.

As mentioned before, our lower bound are based on $\ell_1$ norms, thus we can not transfer to one-way marginals directly. Merely using the properties in the definition of Fingerprinting Codes is not enough for a good lower bound. Instead, we need to make full use of the concrete structure of the codes.

| Article | Constrained? | Loss Function | Pure DP | Approximate DP |
|---|---|---|---|---|
| Bassily et al. (2014) | constrained | GLM | $\Omega(\frac{p}{n\epsilon})$ | $\Omega(\frac{\sqrt{p}}{n\epsilon})$ |
| Song et al. (2021) | unconstrained | GLM | N/A | $\Omega(\frac{\sqrt{\text{rank}}}{n\epsilon})$ |
| Asi et al. (2021) | both | general | N/A | $\Omega(\frac{\sqrt{p}}{n\epsilon \log p})$ |
| Ours | both | general | $\Omega(\frac{p}{n\epsilon})$ | $\Omega(\frac{\sqrt{p \log(1/\delta)}}{n\epsilon})$ |

Table 1: Comparison on lower bounds for private convex ERM. Our lower bounds can be extended to constrained case easily. The lower bound of Song et al. (2021) is weaker than ours in the important $p \gg n$ setting.

As for the upper bounds, the private ERM Wang et al. (2017) and private Stochastic Convex Optimization (SCO) Feldman et al. (2020) for convex and smooth functions are extensively studied, where the objective is to minimize the function $\mathbb{E}_{d \sim \mathcal{P}}[\ell(\theta; d)]$ in the SCO and people only need (nearly) linear gradient queries to get optimal excess loss. But for convex functions without any smoothness assumption, the current best algorithms Kulkarni et al. (2021); Asi et al. (2021) will need more queries ($n^{1.375}$ in the worst case). Besides, most of the previous works are considering problems in $\ell_2$ norm, and there are some recent results Bassily et al. (2021); Asi et al. (2021) studying the general $\ell_p$ norm.

### 1.5 ROADMAP

In section 2 we introduce background knowledge needed in the rest of the paper. In section 3 we prove the main result of this paper, an $\Omega(\min(1, \frac{\sqrt{p \log(1/\delta)}}{n\epsilon}))$ lower bound for approximate DP-ERM in the unconstrained case. In section 4 we discuss an $\Omega(\min(1, \frac{p}{n\epsilon}))$ lower bound for the excess risk of pure DP algorithms for minimizing any unconstrained 1-Lipschitz convex loss function. Section 5 concludes this paper. All missing (technical) proofs can be found in the appendix.

## 2 PRELIMINARY

We consider minimizing the excess risk of unconstrained Lipschiz convex function with DP algorithms in this paper, where we let $n$ denote the sample size and $p$ be the dimension of a sample. In this section, we will introduce main background knowledge required in the rest of the paper. Additional background knowledge such as the definition of GLM can be found in appendix.

**Definition 2.1** (Differential privacy). A randomized mechanism $\mathcal{M}$ is $(\epsilon, \delta)$-differentially private if for any event $\mathcal{O} \in \text{Range}(\mathcal{M})$ and for any neighboring databases $\mathcal{D}$ and $\mathcal{D}'$ that differ in a single data element, one has

$$\Pr[\mathcal{M}(\mathcal{D}) \in \mathcal{O}] \le \exp(\epsilon) \Pr[\mathcal{M}(\mathcal{D}') \in \mathcal{O}] + \delta.$$

When $\delta > 0$, we refer to the above condition as approximate differential privacy. The special case when $\delta = 0$ is called pure differential privacy.

**Definition 2.2** (Empirical Risk Minimization). Given a family of convex loss functions $\{\ell(\theta, d)\}_{d \in \mathcal{D}}$ of $\theta$ over $\mathcal{K} \subseteq \mathbb{R}^p$ and a set of samples $\mathcal{D} = \{d_1, \cdots, d_n\}$ over the universe $\mathcal{D}$, the objective of Empirical Risk Minimization (ERM) is to minimize

$$L(\theta; \mathcal{D}) = \frac{1}{n} \sum_{i=1}^{n} \ell(\theta; d_i).$$

The excess empirical loss with respect to a solution $\theta$ is defined by

$$L(\theta; \mathcal{D}) - L(\theta^*; \mathcal{D})$$

where $\theta^* \in \arg\min_{\theta \in \mathcal{K}} L(\theta; \mathcal{D})$, measuring the performance of the solution $\theta$ compared with the best solution in $\mathcal{K}$.

**Definition 2.3** (G-Lipschitz Continuity). A function $f : \mathbb{R}^p \to \mathbb{R}$ is $G$-Lipschitz continuous with respect to $\ell_2$ norm if the following holds for all $\theta, \theta' \in \mathbb{R}^p$:

$$|f(\theta) - f(\theta')| \le G\|\theta - \theta'\|_2 \tag{1}$$

The Chernoff Bound will serve to prove the fingerprinting code constructed in Bun et al. (2018) satisfies our modified definition of fingerprinting code as well.

**Proposition 2.4** (The Chernoff Bound). *Let $X = \sum_{i=1}^{n} X_i$ where $X_i = 1$ with probability $p_i$ and $X_i = 0$ with probability $1 - p_i$. Assume all $X_i$ are independent random variables. Let $u = \sum_{i=1}^{n} p_i$. Then*

$$P(|X - u| \geq \delta u) \leq 2 exp(-u\delta^2/2). \tag{2}$$

## 3 APPROXIMATE DP

In this section, we consider the lower bound for approximate differential privacy where $2^{-O(n)} < \delta < o(1/n)$. Such assumption on $\delta$ is common in literature, for example in Steinke and Ullman (2015). We briefly introduce the (classic) fingerprinting codes first:

### 3.1 FINGERPRINTING CODES

**Definition 3.1** (Fingerprinting codes). We are given $n, p \in \mathbb{N}, \xi \in (0, 1]$. A pair of (random) algorithms (Gen, Trace) is called an $(n, p)$-fingerprinting code with security $\xi \in (0, 1]$ if Gen outputs a code-book $C \in \{0, 1\}^{n \times p}$ and for any (possibly randomized) adversary $\mathcal{A}_{FP}$ and any subset $S \subseteq [n]$, if we set $c \leftarrow_R \mathcal{A}_{FP}(C_S)$, then

- $\Pr[c \in F(C_S) \bigwedge \text{Trace}(C, c) = \perp] \leq \xi$

- $\Pr[\text{Trace}(C, c) \in [n] \backslash S] \leq \xi$

where $F(C_S) = \{c \in \{0, 1\}^d \mid \forall j \in [d], \exists i \in S, c_j = c_{ij}\}$, and the probability is taken over the coins of Gen, Trace and $\mathcal{A}_{FP}$.

There is a very good motivation behind the fingerprinting codes. For example, a software distributor adds a fingerprint to each copy of her software to protect the IP. A coalition of malicious users can compare their copies and find the digits that differ which belong to the fingerprint. For other locations they can't decide and won't change them, which is called the marking condition. This is the reason that we requires $c \in F(C_S)$.

The two properties of fingerprinting codes demonstrate that one can identify at least one malicious user among all with high probability. Bun et al. (2018) extends the definition that the codes can tolerate a small fraction of errors in the marking condition. We further modify this definition, requiring the codes to have biased means, see below.

### 3.2 OUR RESULT

We modify the definition of fingerprinting code instead for our analysis.

**Definition 3.2** (Error Robust Biased Mean Fingerprinting Codes). Given $n, p \in \mathbb{N}, \xi, \beta, \alpha_1, \alpha_2, \alpha_3 \in (0, 1]$. We say a pair of (random) algorithms (Gen, Trace) is an $(n, p)$-fingerprinting code with security $\xi$ and $(\alpha_1, \alpha_2, \alpha_3)$-biased mean, robust to a $\beta$ fraction of errors if Gen outputs a code-book $C \in \{0, 1\}^{n \times p}$ and for any (possibly randomized) adversary $\mathcal{A}_{FP}$ and any coalition $S \subseteq [n]$, if we set $c \leftarrow_R \mathcal{A}_{FP}(C_S)$, then

- $\Pr[c \in F_\beta(C_S) \bigwedge \text{Trace}(C, c) = \perp] \leq \xi$

- $\Pr[\text{Trace}(C, c) \in [n] \backslash S] \leq \xi$

- $\Pr[G_{\alpha_1}(C) \geq (1 - \alpha_2)] \leq \alpha_3$

where $F_\beta(C_S) = \{c \in \{0, 1\}^p \mid Pr_{j \leftarrow_R [p]}[\exists i \in S, c_j = c_{ij}] \geq 1 - \beta\}$, $G_\alpha(C_S) = |\{j : |\sum_{i \in S} c_{ij}/|S| - 1/2| \leq \alpha\}|$ is the number of slightly biased columns in $C_S$ and the probability is taken over the coins of Gen, Trace and $\mathcal{A}_{FP}$.

We use the fingerprinting code in Bun et al. (2018) for the construction of our lower bound, see Algorithm 1 in the appendix. We utilize an $\ell_1$ loss and use the fingerprinting code in Bun et al.

(2018) as our 'hard case'. To proceed, we first introduce a few lemmas which would be of use later. Similar to Bun et al. (2018), we have the following standard lemma which allows us to reduce any $\epsilon < 1$ to $\epsilon = 1$ case without loss of generality, using the well-known 'secrecy of the sample' lemma from Kasiviswanathan et al. (2011).

**Lemma 3.1.** *A condition Q has sample complexity $n^*$ for algorithms with $(1, o(1/n))$-differential privacy ($n^*$ is the smallest sample size that there exists an $(1, o(1/n))$-differentially private algorithm $\mathcal{A}$ which satisfies Q), if and only if it also has sample complexity $\Theta(n^*/\epsilon)$ for algorithms with $(\epsilon, o(1/n))$-differential privacy.*

Notice that Lemma 3.1 discusses the sample complexity of the algorithm, therefore is independent of the $(\alpha_1, \alpha_2, \alpha_3)$-biased mean appeared in the above definition which only concerns the construction of the fingerprinting code. The following lemma verifies that the fingerprinting code Algorithm 1 indeed has biased mean as in definition 3.2. The proof is straightforward by using the Chernoff bound multiple times.

**Lemma 3.2.** *Algorithm 1 (the fingerprinting code) has $(1/100, 999/1000, \exp(-\Omega(p)))$-biased mean.*

Directly combining Lemma 3.2 and Theorem 3.4 from Bun et al. (2018), we have the following lemma, which states that for the fingerprinting code Algorithm 1 which we will use in proving our main theorem to satisfy the error robust biased mean property in definition 3.2, one needs roughly $\tilde{\Omega}(\sqrt{p})$ samples.

**Lemma 3.3.** *For every $p \in \mathbb{N}$ and $\xi \in (0, 1]$, there exists an $(n, p)$-fingerprinting code (Algorithm 1) with security $\xi$ and $(1/100, 999/1000, \exp(-\Omega(p)))$-biased mean, robust to a $1/75$ fraction of error for*

$$n = n(p, \xi) = \tilde{\Omega}(\sqrt{p/\log(1/\xi)}).$$

We are ready to prove the main result of this section by using Lemma 3.3 to reach a contradiction. Consider the following $\ell_1$ norm loss function. Define

$$\ell(\theta; d) = ||\theta - d||_1, \theta, d \in \mathbb{R}^p \tag{3}$$

For any data-set $\mathcal{D} = \{d_1, ..., d_n\}$, we define $L(\theta; \mathcal{D}) = \frac{1}{n} \sum_{i=1}^{n} \ell(\theta; d_i)$.

**Theorem 3.4** (Lower bound for $(\epsilon, \delta)$-differentially private algorithms). *Let $n, p$ be large enough and $1 \geq \epsilon > 0, 2^{-O(n)} < \delta < o(1/n)$. For every $(\epsilon, \delta)$-differentially private algorithm with output $\theta^{priv} \in \mathbb{R}^p$, there is a data-set $\mathcal{D} = \{d_1, ..., d_n\} \subset \{0, 1\}^p \cup \{\frac{1}{2}\}^p$ such that*

$$\mathbb{E}[L(\theta^{priv}; \mathcal{D}) - L(\theta^\star; \mathcal{D})] = \Omega(\min(1, \frac{\sqrt{p \log(1/\delta)}}{n\epsilon})GC) \tag{4}$$

*where $\ell$ is G-Lipschitz, $\theta^\star$ is a minimizer of $L(\theta; \mathcal{D})$, and $C$ is the diameter of the set $\{\arg \min_\theta L(\theta; \mathcal{D}) | \mathcal{D} \subset \{0, 1\}^{n \times p}\}$, which contains all possible true minimizers.*

Due to the space limit, we leave the proof of the Theorem 3.4 in the appendix.

The dependence on the diameter $C$ makes sense as one can minimize a substitute loss function $\ell'(x) = \ell(ax)$ where $a \in (0, 1)$ is a constant instead, which decreases Lipschitz constant $G$ but increases the diameter $C$. Note also that $C > 0$ whenever all possible $\mathcal{D}$ don't share the same minimizer of $L$, which is often the case. This bound improves a log factor over Bassily et al. (2014) by combining the the group privacy technique in Steinke and Ullman (2015) and our modified definition of fingerprinting code.

We leave several remarks discussing slight generalizations of Theorem 3.4.

**Remark 3.5.** *Our lower bound can be directly extended to the constrained setting, by setting the constrained domain to be $[0, 1]^{n \times p}$ which contains the convex hull of all possible minimizers $\{\arg \min_\theta L(\theta; \mathcal{D}) | \mathcal{D} \subset \{0, 1\}^{n \times p}\}$.*

**Remark 3.6.** *Similarly, we can derive an $\Omega(\min(1, \frac{\sqrt{\text{rank} \log(1/\delta)}}{n\epsilon}))$ lower bound when we additionally assume the rank of gradient subspace. The analysis remains the same except we first apply orthogonal transformation then set the complement of the gradient subspace to be all 0's in $\mathcal{D}$.*

**Remark 3.7.** *The third property of definition 3.2 serves the group privacy analysis to further improve a $\log(1/\delta)$ term over Bassily et al. (2014). One can simplify the proof by setting $k = 1$ and borrow the lower bound for 1-way marginals from Bun et al. (2018), at the cost of losing this $log(1/\delta)$ term. See appendix for details.*

## 4 PURE DP

In this section, we give a lower bound for $\epsilon$-(pure) differentially private algorithms for minimizing unconstrained convex Lipschitz loss function $L(\theta; \mathcal{D})$. In the construction of lower bounds for constrained DP-ERM (Bassily et al. (2014)), they chose linear function $\ell(\theta; d) = \langle \theta, d \rangle$ as their objective function which isn't applicable in the unconstrained setting because it could decrease to negative infinity. Instead, we use a novel $\ell_2$ norm loss function to over come this problem:

$$\ell(\theta; d) = ||\theta - d||_2, \theta, d \in \mathbb{R}^p \tag{5}$$

For any dataset $\mathcal{D} = \{d_1, ..., d_n\}$, we define $L(\theta; \mathcal{D}) = \frac{1}{n} \sum_{i=1}^{n} \ell(\theta; d_i)$. Clearly, both $\ell$ and $L$ are convex and 1-Lipschitz. The structure of the proof is similar to that in Bassily et al. (2014), while technical details are quite different as we need to handle a non-linear objective function. Different from the simple average of points in Bassily et al. (2014), we need to consider the Fermat point instead, which is the minimizer of the $\ell_2$ norm loss function.

### 4.1 FERMAT POINT

**Definition 4.1** (Fermat point). *The set of Fermat points $P(\mathcal{D})$ of a dataset $\mathcal{D} = \{d_1, ..., d_n\}$ contains points minimizing its $\ell_2$ distance to all points in $\mathcal{D}$:*

$$P(\mathcal{D}) = \{\arg \min_{x \in R^p} \sum_{i=1}^{n} ||x - d_i||_2\} \tag{6}$$

One obstacle of using $\ell_2$ norm as our loss is that Fermat points aren't unique in the worst case. Given a (finite) dataset $\mathcal{D}$, we can easily see that $P(\mathcal{D})$ is a compact subset of the convex hull of $\mathcal{D}$, which encourages us to define a unique "maximum" element in $P(\mathcal{D})$. To do so, we introduce the following well-order on $\mathbb{R}^p$.

**Definition 4.2** (Coordinate dictionary order). *A point $x$ is said to be larger than $y$ in coordinate dictionary order if and only if there exists an index $i \in [n]$ such that $x_i > y_i$, and for any $j < i$ we have that $x_j = y_j$.*

It's straightforward to verify that CDO (coordinate dictionary order) is a well-order. Next we use CDO to select a unique member from the set $P(\mathcal{D})$ of all Fermat points.

**Definition 4.3** (Ordered Fermat point). *The Ordered Fermat point $q(\mathcal{D})$ of a dataset $\mathcal{D} = \{d_1, ..., d_n\}$ is defined as:*

$$q(\mathcal{D}) = \arg \max_{x \in P(\mathcal{D})} \text{CDO}(x) \tag{7}$$

Such $q(\mathcal{D})$ must exist for a finite dataset as long as $P(\mathcal{D})$ is compact and non-empty, because there can't be an ordered infinite sequence with its limit outside of $P(\mathcal{D})$ which contradicts compactness. The technical proof of the following proposition is deferred to appendix.

**Proposition 4.1.** $q(\mathcal{D})$ *always exists for a finite dataset $\mathcal{D}$.*

Note that $q(\mathcal{D})$ is unique by definition and is always a minimizer of $L(\theta; \mathcal{D})$ over $\mathbb{R}^p$. In the following subsection we are going to show that any pure DP algorithm can't estimate $q(\mathcal{D})$ with good accuracy, then prove that a large error in estimating $q(\mathcal{D})$ will lead to large error in the excess risk of $\ell_2$ norm loss as well, establishing the main lower bound of this section.

### 4.2 LOWER BOUND

In this subsection, we prove a lower bound on the excess risk incurred by any $\epsilon$-differentially private algorithm whose output is denoted by $\theta^{priv} \in \mathbb{R}^p$. We first introduce the following lemma showing

that it's impossible to find the location of the ordered Fermat point $q(\mathcal{D})$ with good accuracy using a pure DP algorithm.

The proof follows the spirit of Bassily et al. (2014), constructing datasets 'far away' from each other such that the events of estimating the Fermat point of each dataset accurately are mutually disjoint. Then by differential privacy as long as one can estimate one dataset accurately, one can estimate any other one with certain probability as well. The sum of all these probabilities is no more than 1 due to the disjointness, which leads to the desired bound.

We denote $e_1 \triangleq (1, 0, ..., 0)^\top$ and let $\oplus$ denote the direct sum of vectors, i.e. $\alpha \oplus \beta = (\alpha, \beta)$ where $\alpha \in \mathbb{R}^a, \beta \in \mathbb{R}^b$ are both vectors. For a vector $\alpha$ and a set $S$, we denote $\alpha \oplus S = \{(\alpha, \beta) : \beta \in S\}$.

**Lemma 4.4.** *Let $n, p \geq 2$ and $\epsilon > 0$. There is a number $M = \Omega(\min(n, \frac{p}{\epsilon}))$ such that for any $\epsilon$-differentially private algorithm $A$, there is a dataset $\mathcal{D} = \{d_1, ..., d_n\} \subset \left(0 \oplus \{\frac{1}{\sqrt{p-1}}, -\frac{1}{\sqrt{p-1}}\}^{p-1}\right) \cup \{e_1, -e_1, 0\}$ with $\|\sum_{i=1}^{n} d_i\|_2 \leq M$ such that, with probability at least $1/2$ (taken over the algorithm random coins), we have*

$$\|A(\mathcal{D}) - q(\mathcal{D})\|_2 = \Omega(\min(1, \frac{p}{n\epsilon})) \tag{8}$$

The classic analysis of Bassily et al. (2014) contains an 'adding dummy points' which will perturb the location of the minimizer. In the constrained case, such perturbation won't change the direction of the minimizer (seen as a vector), but in the unconstrained case non-linear loss functions no longer enjoy such good properties. To oversome this issue, we introduce the auxiliary dimension and the dummy points we add have support only in this dimension. The benefit of doing so is that the Fermat point $q(\mathcal{D})$ will also only change along its direction after we add dummy points, which simplifies the computation.

Lemma 4.4 implies that it's impossible to estimate the ordered Fermat point with good accuracy using a pure DP algorithm. In the following theorem we are going to show that a bad estimate on the ordered Fermat point leads to higher $\ell_2$ norm loss. As the fermat point is a minimizer of $\ell_2$ norm loss, we can naturally translate the discrepancy in estimating $q(\mathcal{D})$ to the excess risk.

**Theorem 4.5** (Lower bound for $\epsilon$-differentially private algorithms). *Let $n, p \geq 2$ and $\epsilon > 0$. For every $\epsilon$-differentially private algorithm with output $\theta^{priv} \in \mathbb{R}^p$, there is a dataset $\mathcal{D} = \{d_1, ..., d_n\} \subset \left(0 \oplus \{\frac{1}{\sqrt{p-1}}, -\frac{1}{\sqrt{p-1}}\}^{p-1}\right) \cup \{e_1, -e_1, 0\}$ such that, with probability at least $1/2$ (over the algorithm random coins), we must have that*

$$L(\theta^{priv}; \mathcal{D}) - \min_\theta L(\theta; \mathcal{D}) = \Omega(\min(1, \frac{p}{n\epsilon})) \tag{9}$$

The proof is based on calculation in the two-dimensional subspace spanned by $q(\mathcal{D})$ and the auxiliary dimension. By observing that $q(\mathcal{D})$ is perpendicular to the auxiliary dimension, we can parameterize $\theta^{priv}$ by these two unit vectors and write down the expression of $L(\theta^{priv}; \mathcal{D}) - \min_\theta L(\theta; \mathcal{D})$ explicitly. Then by elementary inequality scaling we get the desired result.

**Remark 4.6.** *In fact the lower bound in Theorem 4.5 also holds for the case $p = 1$. The only difference is that the case $p = 1$ doesn't need the auxiliary dimension because the perturbation of the minimizer is always along its direction. We can simply use dummy points $\{1, -1, 0\}$ and a similar analysis to Bassily et al. (2014) to achieve this result.*

## 5 CONCLUSION

In this paper, we study differentially private convex ERM in the unconstrained case and give the first tight lower bounds for approximate-DP ERM for general loss functions. Our results also directly imply a same lower bound for the constrained case, improving the classic lower bound in Bassily et al. (2014) by $\log(1/\delta)$. We also give an $\Omega(\frac{p}{n\epsilon})$ lower bound for unconstrained pure-DP ERM which recovers the result in the constrained case. Our techniques enrich the quite limited tools in constructing lower bounds in the private setting and we hope they can find future use, especially for those problems which are not (easily) reducible from one-way marginals. Designing better algorithms for general (un)constrained DP-ERM based on our insights would also be an interesting and meaningful direction, which we leave as future work.

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

# A  ADDITIONAL BACKGROUND KNOWLEDGE

## A.1  GENERALIZED LINEAR MODEL (GLM)

The generalized linear model (GLM) is a flexible generalization of ordinary linear regression that allows for response variables that have error distribution models other than a normal distribution. To be specific,

**Definition A.1** (Generalized linear model (GLM)). *The generalized linear model (GLM) is a special class of ERM problems where the loss function $\ell(\theta, d)$ takes the following inner-product form:*

$$\ell(\theta; d) = \ell(\langle \theta, x \rangle; y) \tag{10}$$

*for $d = (x, y)$. Here, $x \in \mathbb{R}^p$ is usually called the feature vector and $y \in \mathbb{R}$ is called the response.*

## A.2  PROPERTIES OF DIFFERENTIAL PRIVACY

In this subsection we introduce several very basic properties of differential privacy without proving them (refer Dwork et al. (2014) for details). Readers familiar with the field of differential privacy can feel free to skip this section.

**Proposition A.1** (Group privacy). *If $\mathcal{M} : X^n \to Y$ is $(\epsilon, \delta)$-differentially private mechanism, then for all pairs of datasets $x, x' \in X^n$, then $\mathcal{M}(x), \mathcal{M}(x')$ are $(k\epsilon, k\delta e^{k\epsilon})$-indistinguishable when $x, x'$ differs on exact $k$ locations.*

**Proposition A.2** (Post processing). *If $\mathcal{M} : X^n \to Y$ is $(\epsilon, \delta)$-differentially private and $\mathcal{A} : Y \to Z$ is any randomized function, then $\mathcal{A} \circ \mathcal{M} : X^n \to Z$ is also $(\epsilon, \delta)$-differentially private.*

**Proposition A.3** (Composition). *Let $\mathcal{M}_i$ be an $(\epsilon_i, \delta_i)$-differentially private mechanism for all $i \in [k]$. If $\mathcal{M}_{[k]}$ is defined to be*

$$\mathcal{M}_{[k]}(x) = (\mathcal{M}_1(x), ..., \mathcal{M}_k(x)) \tag{11}$$

*then $\mathcal{M}_{[k]}$ is $(\sum_{i=1}^{k} \epsilon_i, \sum_{i=1}^{k} \delta_i)$-differentially private.*

# B  FINGERPRINTING CODE

In this section we briefly introduce the mechanism of the fingerprinting code Algorithm 1. The sub-procedure part is the original fingerprinting code in Tardos (2008), with a pair of randomized algorithms (Gen, Trace). The code generator Gen outputs a codebook $C \in \{0, 1\}^{n \times p}$. The $ith$ row of $C$ is the codeword of user $i$. The parameter $p$ is called the length of the fingerprinting code.

The security property of fingerprinting codes asserts that any codeword can be "traced" to a user $i$. Moreover, we require that the fingerprinting code can find one of the malicious users even when they get together and combine their codewords in any way that respects the marking condition. That is, there is a tracing algorithm Trace that takes as inputs the codebook $C$ and the combined codeword $c'$ and outputs one of the malicious users with high probability.

The sub-procedure Gen' first uses a $\sin^2 x$ like distribution to generate a parameter $p_j$ (the mean) for each column $j$ independently, then generates $C$ randomly by setting each element to be 1 with probability $p_j$ according to its location. The sub-procedure Trace' computes a threshold value $Z$ and a 'score function' $S_i(c')$ for each user $i$, then report $i$ when its score is higher than the threshold.

The main-procedure was introduced in Bun et al. (2018), where Gen adds dummy columns to the original fingerprinting code and applies a random permutation. Trace can first 'undo' the permutation and remove the dummy columns, then use Trace' as a black box. This procedure makes the fingerprinting code more robust in that it tolerates a small fraction of errors to the marking condition.

# C  OMITTED PROOFS

## C.1  PROOF OF LEMMA 3.1

*Proof.* The proof uses a black-box reduction, therefore doesn't depend on $Q$. The direction that $O(n^*/\epsilon)$ samples are sufficient is equal to proving the assertion that given a $(1, o(1/n))$-

---

**Algorithm 1** The Fingerprinting Code (Gen, Trace)

1: **Sub-procedure** $\text{Gen}'$:
2: Let $d = 100n^2 \log(n/\xi)$ be the length of the code.
3: Let $t = 1/300n$ be a parameter and let $t'$ be such that $sin^2 t' = t$.
4: **for** $j = 1, ..., d$: **do**
5:     Choose random $r$ uniformly from $[t', \pi/2 - t']$ and let $p_j = sin^2 r_j$. Note that $p_j \in [t, 1-t]$.

6:     For each $i = 1, ..., n$, set $C_{ij} = 1$ with probability $p_j$ independently.
7: **end for**
8: **return** $C$
9: **Sub-procedure** $\text{Trace}'(C, c')$:
10: Let $Z = 20n \log(n/\xi)$ be a parameter.
11: For each $j = 1, ..., d$, let $q_j = \sqrt{(1 - p_j)/p_j}$.
12: For each $j = 1, ..., d$, and each $i = 1, ..., n$, let $U_{ij} = q_j$ if $C_{ij} = 1$ and $U_{ij} = -1/q_j$ else wise.

13: **for** each $i = 1, ..., n$: **do**
14:     Let $S_i(c') = \sum_{j=1}^{d} c'_j U_{ij}$
15:     Output $i$ if $S_i(c') \geq Z/2$.
16:     Output $\perp$ if $S_i(c') < Z/2$ for every $i = 1, ..., n$.
17: **end for**
18: **Main-procedure** Gen:
19: Let $C$ be the (random) output of $\text{Gen}'$, $C \in \{0, 1\}^{n \times d}$
20: Append $2d$ 0-marked columns and $2d$ 1-marked columns to $C$.
21: Apply a random permutation $\pi$ to the columns of the augmented codebook.
22: Let the new codebook be $C' \in \{0, 1\}^{n \times 5d}$.
23: **return** $C'$
24: **Main-procedure** $\text{Trace}(C, c')$:
25: Obtain $C'$ from the shared state with Gen.
26: Obtain $C$ by applying $\pi^{-1}$ to the columns of $C'$ and removing the dummy columns.
27: Obtain $c$ by applying $\pi^{-1}$ to $c'$ and removing the symbols corresponding to fake columns.
28: **return** $i$ randomly from $\text{Trace}'(C, c)$.

---

differentially private algorithm $\mathcal{A}$, we can get a new algorithm $\mathcal{A}'$ with $(\epsilon, o(1/n))$-differential privacy at the cost of shrinking the size of the dataset by a factor of $\epsilon$.

Given input $\epsilon$ and a dataset $X$, we construct $A'$ to first generate a new dataset $T$ by selecting each element of $X$ with probability $\epsilon$ independently, then feed $T$ to $\mathcal{A}$. Fix an event $S$ and two neighboring datasets $X_1, X_2$ that differs by a single element $i$. Consider running $\mathcal{A}$ on $X_1$. If $i$ is not included in the sample $T$, then the output is distributed the same as a run on $X_2$. On the other hand, if $i$ is included in the sample $T$, then the behavior of $\mathcal{A}$ on $T$ is only a factor of $e$ off from the behavior of $\mathcal{A}$ on $T \setminus \{i\}$. Again, because of independence, the distribution of $T \setminus \{i\}$ is the same as the distribution of $T$ conditioned on the omission of $i$.

For a set $X$, let $p_X$ denote the distribution of $\mathcal{A}(X)$, we have that for any event $S$,

$$p_{X_1}(S) = (1 - \epsilon)p_{X_1}(S|i \notin T) + \epsilon p_{X_1}(S|i \in T)$$
$$\leq (1 - \epsilon)p_{X_2}(S) + \epsilon(e \cdot p_{X_2}(S) + \delta)$$
$$\leq \exp(2\epsilon)p_{X_2}(S) + \epsilon\delta$$

A lower bound of $p_{X_1}(S) \geq \exp(-\epsilon)p_{X_2}(S) - \epsilon\delta/e$ can be obtained similarly. To conclude, since $\epsilon\delta = o(1/n)$ as the sample size $n$ decreases by a factor of $\epsilon$, $\mathcal{A}'$ has $(2\epsilon, o(1/n))$-differential privacy. The size of $X$ is roughly $1/\epsilon$ times larger than $T$, combined with the fact that $\mathcal{A}$ has sample complexity $n^*$ and $T$ is fed to $\mathcal{A}$, $\mathcal{A}'$ has sample complexity at least $\Theta(n^*/\epsilon)$.

For the other direction, simply using the composability of differential privacy yields the desired result. In particular, by the $k$-fold adaptive composition theorem in Dwork et al. (2006), we can combine $1/\epsilon$ independent copies of $(\epsilon, \delta)$-differentially private algorithms to get an $(1, \delta/\epsilon)$ one and notice that if $\delta = o(1/n)$, then $\delta/\epsilon = o(1/n)$ as well because the sample size $n$ is scaled by a factor of $\epsilon$ at the same time, offsetting the increase in $\delta$. $\qquad\square$

## C.2 PROOF OF LEMMA 3.2

*Proof.* In line 5 of algorithm 1, every column $j$ is assigned a probability $p_j$ independently where

$$Pr[|p_j - \frac{1}{2}| < 0.002] < \frac{1}{400} \tag{12}$$

by straightforward calculation. By the Chernoff bound (with $u < p/400, \delta = 1$), with probability at least

$$1 - 2\exp(-p/800) \tag{13}$$

, at least $1 - \frac{1}{200}$ fraction of the columns have $|p_j - \frac{1}{2}| \geq 0.002$. Denote $m_j$ to be the mean of entries of column $j$, then by using the Chernoff bound again (with $\delta = 0.001$), we have that with probability at least

$$1 - 2\exp(-n/8000000) \tag{14}$$

a column $j$ actually satisfies $|m_j - \frac{1}{2}| \geq 0.001$. Again by the Chernoff bound (with $u \leq 2\exp(-n/8000000)p$ and $u\delta = 0.01p$) together with the union bound, at least 0.99 fraction of all columns have $|m_j - \frac{1}{2}| \geq 0.001$ with probability at least

$$1 - 2\exp(-p/800) - 2\exp(-pe^{n/8000000}/40000) = 1 - O(e^{-\Omega(p)}) \tag{15}$$

$\square$

## C.3 PROOF OF THEOREM 3.4

*Proof.* Let $(\alpha_1, \alpha_2, \alpha_3) = (1/100, 999/1000, \exp(-\Omega(p)))$ be the parameters in the statement of Lemma 3.3. Let $k = \Theta(\log(1/\delta))$ be a parameter to be determined later and $n_k = \lfloor n/k \rfloor$.

Consider the case when $p \geq p_{n_k}$ first, where $p_{n_k} = O(\epsilon^2 n_k^2 \log(1/\delta))$. Without loss of generality, we assume $\epsilon = 1$ first, and $p_{n_k} = O(n_k^2 \log(1/\delta))$ corresponds to the number in Lemma 3.3 where we set $\xi = \delta$. We will use contradiction to prove that for any $(\epsilon, \delta)$-differentially private mechanism $\mathcal{M}$, there exists some $\mathcal{D} \in \{0, 1\}^{n \times p}$ with $G_{\alpha_1 - 1/k}(\mathcal{D}) \leq 1 - \alpha_2$ such that

$$\mathbb{E}[L(\mathcal{M}(\mathcal{D}); \mathcal{D}) - L(\theta^\star; \mathcal{D})] \geq \Omega(p) \tag{16}$$

Assume for contradiction that $\mathcal{M} : \{0, 1\}^{n \times p} \to [0, 1]^{n \times p}$ is a (randomized) $(\epsilon, \delta)$-differentially private mechanism such that

$$\mathbb{E}[L(\mathcal{M}(\mathcal{D}); \mathcal{D}) - L(\theta^\star; \mathcal{D})] < \frac{\alpha_1 \alpha_2 p}{1000}$$

for all $\mathcal{D} \in \{0, 1\}^{n \times p}$ with $G_{\alpha_1 - 1/k}(\mathcal{D}) \leq (1 - \alpha_2)$. We then construct a mechanism $\mathcal{M}_k = \{0, 1\}^{n_k \times p}$ with respect to $\mathcal{M}$ as follows: with input $\mathcal{D}^k \in \{0, 1\}^{n_k \times p}$, $\mathcal{M}_k$ will copy $\mathcal{D}^k$ for $k$ times and append enough 0's to get a dataset $\mathcal{D} \in \{0, 1\}^{n \times p}$. The output is $\mathcal{M}_k(\mathcal{D}^k) = \mathcal{M}(\mathcal{D})$. $\mathcal{M}_k$ is $(k, \frac{e^k - 1}{e - 1}\delta)$-differentially private by the group privacy. According to the construction above, we know that if $G_{\alpha_1}(\mathcal{D}^k) < 1 - \alpha_2$, then $G_{\alpha_1 - 1/k}(\mathcal{D}) < 1 - \alpha_2$ as well.

We consider algorithm $\mathcal{A}_{FP}$ to be the adversarial algorithm in the fingerprinting codes, which rounds the the output $\mathcal{M}_k(\mathcal{D}^k)$ to the binary vector, i.e. rounding those coordinates with values no less than 1/2 to 1 and the remaining 0, and let $c = \mathcal{A}_{FP}(\mathcal{M}(\mathcal{D}))$ be the vector after rounding. As $\mathcal{M}_k$ is $(k, \frac{e^k - 1}{e - 1}\delta)$-differentially private, $\mathcal{A}_{FP}$ is also $(k, \frac{e^k - 1}{e - 1}\delta)$-differentially private.

If for some $\mathcal{D}^k \in \{0, 1\}^{n_k \times p}$ with $G_{\alpha_1}(\mathcal{D}^k) \leq 1 - \alpha_2$, $\mathcal{D}$ (constructed from $\mathcal{D}^k$ as above) further satisfies

$$\mathbb{E}[L(\mathcal{M}(\mathcal{D}); \mathcal{D}) - L(\theta^\star; \mathcal{D})] < \frac{\alpha_1 \alpha_2 p}{1000}.$$

As we are considering the $\ell_1$ loss, we can consider the loss caused by each coordinate independently. Recall that $\mathcal{M}_k(\mathcal{D}^k) = \mathcal{M}(\mathcal{D})$. The fraction of those nearly unbiased columns (the mean is close to 1/2) is at most $1 - \alpha_2$, and we treat the worst-case error for them. For other '$\alpha_1$-biased' columns (coordinates) which take at least $\alpha_2$ fraction of all, if $\mathcal{M}(\mathcal{D})$ is right for the prediction, then appending 0's can't change the prediction and $\mathcal{M}_k(\mathcal{D}_k)$ is also right. Thus we have that

$$\mathbb{E}[L(\mathcal{M}_k(\mathcal{D}^k); \mathcal{D}^k) - L(\theta^\star; \mathcal{D}^k)] < \mathbb{E}[L(\mathcal{M}(\mathcal{D}); \mathcal{D}) - L(\theta^\star; \mathcal{D})] + (1 - \alpha_2)p < \frac{p}{900}.$$

By Markov Inequality we know that

$$\Pr[L(\mathcal{M}_k(\mathcal{D}^k); \mathcal{D}^k) - L(\theta^\star; \mathcal{D}^k)] \geq \frac{p}{180}] \leq 1/5.$$

$c \notin F_\beta(D^k)$ means that there is at least $\beta p$ all-one or all-zero columns in $D^k$, but $c$ is inconsistent in those coordinates. Thus if $c \notin F_\beta(\mathcal{D}^k)$, we have that $L(\mathcal{M}_k(\mathcal{D}^k); \mathcal{D}^k) - L(\theta^\star; \mathcal{D}^k) \geq \beta p/2 = p/150 > p/180$ for the $\mathcal{D}^k$ by Lemma 3.3, implying

$$\Pr[c \in F_\beta(\mathcal{D}^k)] \geq 4/5. \tag{17}$$

By the first property of the codes, one also has

$$\Pr[L(\mathcal{M}_k(\mathcal{D}^k); \mathcal{D}^k) - L(\theta^\star; \mathcal{D}^k) \leq p/180 \bigwedge \text{Trace}(\mathcal{D}^k, c) = \perp]$$

$$\leq \Pr[c \in F_\beta(\mathcal{D}^k) \bigwedge \text{Trace}(\mathcal{D}^k, c) = \perp] \leq \delta.$$

Recall that the arguments above are for those $\mathcal{D}^k \in \{0,1\}^{n_k \times p}$ with $G_{\alpha_1}(\mathcal{D}^k) \leq 1 - \alpha_2$, which happens with probability at least $1 - \alpha_3$ by the third property of fingerprinting codes. By union bound, we can upper bound the probability $\Pr[\text{Trace}(\mathcal{D}^k, c) = \perp] \leq 1/5 + \delta + \alpha_3 \leq 1/2$. As a result, there exists $i^* \in [n_k]$ such that

$$\Pr[i^* \in \text{Trace}(\mathcal{D}^k, c)] \geq 1/(2n_k). \tag{18}$$

Consider the database with $i^*$ removed, denoted by $\mathcal{D}^k_{-i^*}$. Let $c' = \mathcal{A}_{FP}(\mathcal{M}(\mathcal{D}^k_{-i^*}))$ denote the vector after rounding. By the second property of fingerprinting codes, we have that

$$\Pr[i^* \in \text{Trace}(\mathcal{D}^k_{-i^*}, c')] \leq \delta.$$

By the differential privacy and post-processing property of $\mathcal{M}$,

$$\Pr[i^* \in \text{Trace}(\mathcal{D}^k, c)] \leq e^k \Pr[i^* \in \text{Trace}(\mathcal{D}^k_{-i^*}, c')] + \frac{e^k - 1}{e - 1}\delta.$$

which implies that

$$\frac{1}{2n_k} \leq e^{k+1}\delta. \tag{19}$$

Recall that $2^{-O(n)} < \delta < o(1/n)$, and Equation (19) suggests $k/n \leq 2e^k/\delta$ for all valid $k$, but it is easy to see there exists $k = \Theta(\log(1/\delta))$ to make this inequality false, which is contraction. As a result, there exists some $\mathcal{D} \in \{0,1\}^{n \times p}$ with $G_{\alpha_1 - 1/k}(\mathcal{D}) \geq (1 - \alpha_2)$ subject to

$$\mathbb{E}[L(\mathcal{M}(\mathcal{D}); \mathcal{D}) - L(\theta^\star; \mathcal{D})] \geq \frac{\alpha_1 \alpha_2 p}{1000} = \Omega(p).$$

For the $(\epsilon, \delta)$-differential privacy case, setting $Q$ to be the condition

$$\mathbb{E}[L(\mathcal{M}(\mathcal{D}); \mathcal{D}) - L(\theta^\star; \mathcal{D})] = O(p).$$

in Lemma 3.1, we have that any $(\epsilon, \delta)$-differentially private mechanism $\mathcal{M}$ which satisfies $Q$ for all $\mathcal{D} \in \{0,1\}^{n \times p}$ with $G_{\alpha_1 - 1/k}(\mathcal{D}) \geq 1 - \alpha_2$ must have $n \geq \Omega(\sqrt{p \log(1/n)}/\epsilon)$.

Now we consider the case when $p < p_{n_k}$, i.e. when $n > n^\star \triangleq \Omega(\sqrt{p \log(1/\delta)}/\epsilon)$. Given any dataset $\mathcal{D} \in \{0,1\}^{n^\star \times p}$ with $G_{\alpha_1 - 1/k}(\mathcal{D}) \geq 1 - \alpha_2$, we will construct a new dataset $\mathcal{D}'$ based on $\mathcal{D}$ by appending dummy points to $\mathcal{D}$ like in Lemma 4.4. Specifically, if $n - n^\star$ is even, we append $n - n^\star$ rows among which half are 0 and half are $\{1\}^p$. If $n - n^\star$ is odd, we append $\frac{n - n^\star - 1}{2}$ points $0$, $\frac{n - n^\star - 1}{2}$ points $\{1\}^p$ and one point $\{1/2\}^p$.

Denote the new dataset after appending by $\mathcal{D}'$, we will draw contradiction if there is an $(\epsilon, \delta)$-differentially private algorithm $\mathcal{M}'$ such that $\mathbb{E}[L(\mathcal{M}(\mathcal{D}'); \mathcal{D}') - L(\theta^\star; \mathcal{D}')] = o(n^\star p/n)$ for all $\mathcal{D}'$, by reducing $\mathcal{M}'$ to an $(\epsilon, \delta)$-differentially private algorithm $\mathcal{M}$ which satisfies $\mathbb{E}[L(\mathcal{M}(\mathcal{D}); \mathcal{D}) - L(\theta^\star; \mathcal{D})] = o(p)$ for all $\mathcal{D}$ with $G_{\alpha_1 - 1/k}(\mathcal{D}) \geq 1 - \alpha_2$.

We construct $\mathcal{M}$ by first constructing $\mathcal{D}'$, and then use $\mathcal{M}'$ as a black box to get $\mathcal{M}(\mathcal{D}) = \mathcal{M}'(\mathcal{D}')$. It's clear that such algorithm for $\mathcal{D}$ preserves $(\epsilon, \delta)$-differential privacy. It suffices to show that if

$$\mathbb{E}[L(\mathcal{M}'(\mathcal{D}'); \mathcal{D}') - L(\theta^\star; \mathcal{D}')] = o(n^\star p/n), \tag{20}$$

then $L(\mathcal{M}(\mathcal{D}); \mathcal{D}) - L(\theta^\star; \mathcal{D}) = o(p)$, which contradicts the previous conclusion for the case $n \leq n^\star$. Specifically, if $n - n^\star$ is even, we have that

$$n^\star \mathbb{E}[L(\mathcal{M}(\mathcal{D}); \mathcal{D}) - L(\theta^\star; \mathcal{D})] = n\mathbb{E}[L(\mathcal{M}'(\mathcal{D}'); \mathcal{D}') - L(\theta^\star; \mathcal{D}')].$$

and if $n - n^\star$ is odd we have that

$$n^\star \mathbb{E}[L(\mathcal{M}(\mathcal{D}); )\mathcal{D} - L(\theta^\star; \mathcal{D})] \leq n\mathbb{E}[L(\mathcal{M}'(\mathcal{D}'); \mathcal{D}') - L(\theta^\star; \mathcal{D}')] + p/2,$$

both leading to the desired reduction. We try to explain the above two cases in more detail. If $n - n^*$ is even, then the minimizer of $L(; \mathcal{D})$ and $L(\theta^*; \mathcal{D})$ are the same. And the distributions of the $\mathcal{M}(\mathcal{D})$ and $\mathcal{M}'(\mathcal{D}')$ are the same and indistinguishable. Multiplying $n^*$ or $n$ depends on the number of rows (recall that we normalize the objective function in ERM). The second inequality is because we append one point $\{1/2\}^p$, which can only increase the loss $(\|1/2^p - \theta^*\|_1)$ by $p/2$ in the worst case.

Combining results for both cases we have the following:

$$\mathbb{E}[L(\theta^{priv}; \mathcal{D}) - L(\theta^\star; \mathcal{D})] = \Omega(\min(p, \frac{pn^*}{n})) \tag{21}$$

To conclude, observe that $G = \sqrt{p}$ and $C = \sqrt{p}$. In particular, let $\mathcal{D} = (d, ..., d)^\top \in \{0, 1\}^{n \times p}$ contain $n$ identical copies of rows $d \in \{0, 1\}^p$, $\theta^* = d$. Going over all such $\mathcal{D}$, we find that the set $\{\arg\min_\theta L(\theta; \mathcal{D}) | \mathcal{D} \subset \{0, 1\}^{n \times p}\}$ contains $\{0, 1\}^p$, with diameter at least $\sqrt{p}$. Meanwhile, its diameter can't exceed $\sqrt{p}$ obviously. $\qquad\square$

## C.4 DETAILS OF REMARK 3.7

We give a sketch of Remark 3.7. In Bun et al. (2018) they prove the following lower bound for 1-way marginals:

**Proposition C.1** (Corollary 3.6 in Bun et al. (2018)). *The family of 1-way marginals on $\{0, 1\}^d$ has sample complexity at least $\tilde{\Omega}(\sqrt{d})$ for $(1/3, 1/75)$-accuracy and $(O(1), o(1/n))$-differential privacy.*

Inspecting the proof we find that the constant $1/3$ in the above proposition is chosen casually, and can be replaced by any constant $c < 1/2$ for free, as the proof only requires $1 - c$ is rounded to 1 and $c$ is rounded to 0 respectively.

The third property of the fingerprinting code implies that with high probability, at most 0.01 fraction of all columns have mean with bias smaller than 0.001. When we assume the opposite for the sake of contradiction, by union bound, at least $1/75 - 1/100 = 1/300$ fraction of columns have both 'large error on 1-way marginal' and 'large bias on mean'.

For any such column $j$, the algorithm is forced to predict wrongly on the question 'Is there more 0's than 1's in column $j$' as the range of prediction is restricted in $[0, 1]$ and choose $c + 0.001 > 1/2$, thus leading to error on $\ell_1$ norm loss.

## C.5 PROOF OF PROPOSITION 4.1

*Proof.* We assume $P(\mathcal{D}) \neq \emptyset$ without loss of generality. To verify $P(\mathcal{D})$ is compact, we first observe that $P(\mathcal{D})$ is bounded. To prove $P(\mathcal{D})$ is closed, notice that when $P(\mathcal{D}) \neq \emptyset$, the function $f(x) = \sum_{i=1}^n \|x - d_i\|_2$ is continuous and non negative, which implies its image is of the form $[a, \infty)$. Therefore the pre-image of the open set $(a, \infty)$ is also open, whose complement is exactly $P(\mathcal{D})$.

To find the ordered Fermat point, we reduce the dimension of $P(\mathcal{D})$ one after another. Because $P(\mathcal{D})$ is compact, the largest value of the first coordinate $a_1 \triangleq argmax_{x \in P(\mathcal{D})} x_1$ exists, and the ordered Fermat point must lie on the restriction of $P(\mathcal{D})$ on $\{x | x_1 = a_1\}$ which is also compact and non-empty. We continue this process until all dimensions are peeled and there is one point left because the only non-empty set with zero dimension is a single point. $\qquad\square$

## C.6   PROOF OF LEMMA 4.4

*Proof.* By using a standard packing argument we can construct $K = 2^{\frac{p-1}{2}}$ points $d^{(1)}, ..., d^{(K)}$ in $0 \oplus \{\frac{1}{\sqrt{p-1}}, -\frac{1}{\sqrt{p-1}}\}^{p-1}$ such that for every distinct pair $d^{(i)}, d^{(j)}$ of these points, we have

$$||d^{(i)} - d^{(j)}||_2 \geq \frac{1}{8} \tag{22}$$

It is easy to show the existence of such set of points using the probabilistic method (for example, the Gilbert-Varshamov construction of a linear random binary code).

Fix $\epsilon > 0$ and define $n^\star = \frac{p}{160\epsilon}$. Let's first consider the case where $n \leq 4n^\star$. We construct $K$ datasets $\mathcal{D}^{(1)}, ..., \mathcal{D}^{(K)}$ where for each $i \in [K]$, $\mathcal{D}^{(i)}$ contains $n$ copies of $d^{(i)}$. Note that $q(\mathcal{D}^{(i)}) = d^{(i)}$, we have that for all $i \neq j$,

$$||q(\mathcal{D}^{(i)}) - q(\mathcal{D}^{(j)})||_2 \geq \frac{1}{8} \tag{23}$$

Let $A$ be any $\epsilon$-differentially private algorithm. Suppose that for every $\mathcal{D}^{(i)}, i \in [K]$, with probability at least $1/2$, $||A(\mathcal{D}^{(i)}) - q(\mathcal{D}^{(i)})||_2 < \frac{1}{16}$, i.e., $Pr[A(\mathcal{D}^{(i)}) \in B(\mathcal{D}^{(i)})] \geq \frac{1}{2}$ where for any dataset $\mathcal{D}$, $B(\mathcal{D})$ is defined as

$$B(\mathcal{D}) = \{x \in R^p : ||x - q(\mathcal{D})||_2 < \frac{1}{16}\} \tag{24}$$

Note that for all $i \neq j$, $\mathcal{D}^{(i)}$ and $\mathcal{D}^{(j)}$ differs in all their $n$ entries. Since $A$ is $\epsilon$-differentially private, for all $i \in [K]$, we have $Pr[A(\mathcal{D}^{(1)}) \in B(\mathcal{D}^{(i)})] \geq \frac{1}{2}e^{-\epsilon n}$. Since all $B(\mathcal{D}^{(i)})$ are mutually disjoint, then

$$\frac{K}{2}e^{-\epsilon n} \leq \sum_{i=1}^{K} Pr[A(\mathcal{D}^{(1)}) \in B(\mathcal{D}^{(i)})] \leq 1 \tag{25}$$

which implies that $n > 4n^\star$ for sufficiently large $p$, contradicting the fact that $n \leq 4n^\star$. Hence, there must exist a dataset $\mathcal{D}^{(i)}$ on which $A$ makes an $\ell_2$-error on estimating $q(\mathcal{D})$ which is at least $1/16$ with probability at least $1/2$. Note also that the $\ell_2$ norm of the sum of the entries of such $\mathcal{D}^{(i)}$ is $n$.

Next, we consider the case where $n > 4n^\star$. As before, we construct $K = 2^{\frac{p-1}{2}}$ datasets $\tilde{\mathcal{D}}^{(1)}, \cdots, \tilde{\mathcal{D}}^{(K)}$ of size $n$ where for every $i \in [K]$, the first $n^\star$ entries of each dataset $\tilde{\mathcal{D}}^{(i)}$ are the same as dataset $\mathcal{D}^{(i)}$ from before whereas the remaining $n - n^\star$ entries are constructed as follows. The first $\lfloor \frac{n-n^\star}{2} \rfloor$ of those entries are all copies of $e_1$ whereas the following $\lfloor \frac{n-n^\star}{2} \rfloor$ are copies of $-e_1$. The last entry is set to be $0$ when $n - n^\star$ is odd.

Note that any two distinct datasets $\tilde{\mathcal{D}}^{(i)}, \tilde{\mathcal{D}}^{(j)}$ in this collection differ in exactly $n^\star$ entries. Let $A$ be any $\epsilon$-differentially private algorithm for answering $q$. Suppose that for every $i \in [K]$, with probability at least $1/2$, we have that

$$||A(\tilde{\mathcal{D}}^{(i)}) - q(\tilde{\mathcal{D}}^{(i)})||_2 < \frac{n^\star}{32n} \tag{26}$$

Note that for all $i \in [K]$, we have that $q(\tilde{\mathcal{D}}^{(i)}) = \lambda q(\mathcal{D}^{(i)})$ where $\lambda = \frac{n^\star}{\sqrt{n^2 - 2nn^\star}}$ if $n - n^\star$ is even and

$$\lambda = \frac{n^\star - 1}{\sqrt{4\lfloor \frac{n-n^\star}{2} \rfloor^2 - (n^\star - 1)^2}} \tag{27}$$

if $n - n^\star$ is odd. We notice that $\frac{n^\star}{n} \leq \lambda \leq \frac{2n^\star}{n}$, and is independent of the choice of $i$. Now, we define an algorithm $\tilde{A}$ for answering $q$ on datasets $\mathcal{D}$ of size $n^\star$ as follows. First, $\tilde{A}$ computes $\lambda$ and appends $e_1, -e_1, 0$ as above to get a dataset $\tilde{\mathcal{D}}$ of size $n$. Then, it runs $A$ on $\tilde{\mathcal{D}}$ and outputs $\frac{A(\tilde{\mathcal{D}})}{\lambda}$. Hence, by the post-processing propertry of differential privacy, $\tilde{A}$ is $\epsilon$-differentially private since $A$ is $\epsilon$-differentially private. Thus for every $i \in [K]$, with probability at least $1/2$, we have that $||\tilde{A}(\mathcal{D}^{(i)}) - q(\mathcal{D}^{(i)})||_2 < \frac{1}{16}$. However, this contradicts our result in the first part of the proof.

Therefore, there must exist a dataset $\tilde{\mathcal{D}}^{(i)}$ in the above collection such that, with probability at least $1/2$,

$$||A(\tilde{\mathcal{D}}^{(i)}) - q(\tilde{\mathcal{D}}^{(i)})||_2 \geq \frac{n^\star}{32n} \geq \frac{p}{5120\epsilon n} \tag{28}$$

Note that the $\ell_2$ norm of the sum of entries of such $\tilde{D}^{(i)}$ is always $n^\star$. $\qquad\square$

## C.7 PROOF OF THEOREM 4.5

*Proof.* Let $A$ be an $\epsilon$-differentially private algorithm for minimizing $L$ and let $\theta^{priv}$ denote its output. We choose the dataset $\mathcal{D}$ (with corresponding $d_i$) constructed in Lemma 4.4. When $n \leq 4n^\star$, $\mathcal{D}$ contains only identical elements $d_i$ so that $\min_\theta L(\theta; \mathcal{D}) = 0$, and

$$L(\theta^{priv}; \mathcal{D}) - \min_\theta L(\theta; \mathcal{D}) = L(\theta^{priv}; \mathcal{D}) = ||\theta^{priv} - q(\mathcal{D})||_2 = \Omega(\min(1, \frac{p}{n\epsilon})) \tag{29}$$

by Lemma 4.4. When $n > 4n^\star$, we denote $r \triangleq ||\theta^{priv} - q(\mathcal{D})||_2 = \Omega(\min(1, \frac{p}{\epsilon n}))$. Notice that $d_i, e_1, -e_1, 0$ all lie in a 2-dimensional subspace and $||d_i||_2 = 1$ is perpendicular to $e_1$, we may assume $d_i = e_2$ without loss of generality. Because $q(\mathcal{D}) = \lambda e_2$, we parameterize $\theta^{priv}$ as follows

$$\theta^{priv} = (x_1, \lambda + x_2, x_3, ..., x_p) \tag{30}$$

where $\sum_{i=1}^p x_i^2 = r^2$. Now the excess loss satisfies

$L(\theta^{priv}; \mathcal{D}) - \min_\theta L(\theta; \mathcal{D})$

$\geq (n^\star \sqrt{r^2 + (\lambda - 1)^2 + 2(\lambda - 1)x_2} + \lfloor \frac{n - n^\star}{2} \rfloor \sqrt{r^2 + 1 + \lambda^2 + 2\lambda x_2 + 2x_1}$

$+ \lfloor \frac{n - n^\star}{2} \rfloor \sqrt{r^2 + 1 + \lambda^2 + 2\lambda x_2 - 2x_1})/n$

(opening up the expression)

$\geq (\lfloor \frac{n - n^\star}{2} \rfloor \sqrt{r^2 + (\lambda - 1)^2 + 2x_1} + \lfloor \frac{n - n^\star}{2} \rfloor \sqrt{r^2 + (\lambda - 1)^2 - 2x_1})/n$

(dropping the first term)

$\geq (\lfloor \frac{n - n^\star}{2} \rfloor \sqrt{r^2 + (\lambda - 1)^2})/n$

$(\max\{x_1, -x_1\} \geq 0)$

$\geq \lfloor \frac{n - n^\star}{2} \rfloor \cdot \frac{r}{n} = \Omega(\min(1, \frac{p}{n\epsilon}))$

$(n > 4n^\star, r = \Omega(\min(1, \frac{p}{\epsilon n})))$

$\qquad\square$

