# OpenReview forum: "Tight lower bounds for Differentially Private ERM"
_ICLR.cc/2022/Conference — ICLR 2022 Submitted_

### Official Review · Reviewer_bqs4 · 2021-10-25

**Correctness:** 4
**Technical Novelty And Significance:** 3
**Empirical Novelty And Significance:** 1
**Recommendation:** 8
**Confidence:** 3

**Main Review:**

Strengths:
--This paper is the first to give tight lower bounds for approximate DP ERM for general loss functions.

--Its lower bound for the constrained case improves an existing lower bound.

--New techniques were developed as a modification of the regular fingerprinting codes, which are of independent interest as well, especially for those problems that are not easily reducible from one-way marginals.

--The pure DP ERM lower bounds for the unconstrained setting seem tight.

**Summary Of The Paper:**

This paper studies differentially private empirical risk minimisation (ERM) in the unconstrained setting. It gives tight lower bounds for approximate DP ERM for general loss functions, which also implies the same lower bound for the constrained case, which is an improvement over a classic lower bound by Bassily et al 2014. It also gives a lower bound for unconstrained pure DP ERM that recovers the result in the constrained case.

**Summary Of The Review:**

--The work appears technically strong and non-trivial.

--The contributions may be viewed as incremental, but the techniques developed could be useful for other problems as well.

--This is a lower bounds paper, so my score for its empirical contribution is not relevant.

---

> ### Author Response · Authors · 2021-11-15
> **Reply to Reviewer bqs4**
>
> Thanks for your positive feedback!
> If you have any further questions or suggestions, we would be very happy to answer them and polish our work.

---

### Official Review · Reviewer_E1tY · 2021-10-30

**Correctness:** 2
**Technical Novelty And Significance:** 3
**Empirical Novelty And Significance:** Not applicable
**Recommendation:** 5
**Confidence:** 3

**Main Review:**

## Strengths
- ERM is a fundamental (and ubiquitous) problem and this paper essentially closes the gap left in literature for DP ERM.
- The proof requires non-trivial extensions of previous works (e.g. the mean-biased property of fingerprinting codes) which may be useful elsewhere.

## Weaknesses
- The proof of the pure-DP lower bound does not seem correct as the authors drop some terms that should be there. (More details in "Detailed Comments" below.)
- The quantitative improvements for the approximate-DP case is pretty small, i.e. $\sqrt{\log(1/\delta)}$ and furthermore, it seems to be known among the experts that this follows from literature. (E.g. [Song et al., AISTATS 2021] states in footnote 5 that "dependence on the $\delta$ parameter can be introduced into the lower bound by a generic group privacy reduction [Steinke and Ullman, 2015]".) I still think it is helpful to be written down formally but it does question the novelty of the technique a bit.

## Detailed Comments
- *Issue with pure-DP proof*: I do not believe that the proof of Theorem 4.5 in Appendix C.7 is correct. Specifically, in the expression after $L(\\theta^{priv}; \\mathcal{D}) - \\min_{\\theta} L(\\theta; \\mathcal{D})$, you seem to completely drop the second term $\\min_{\\theta} L(\\theta; \\mathcal{D})$, which is *not* equal to zero. (E.g. since $q(\mathcal{D}) = \lambda e_2$, clearly the loss w.r.t. the points $e_1$ is non-zero.) It is not clear to me whether the proof would work if this term is not incorrectly drop.

### Other More Minor Comments
- I do find the entire discussion about uniqueness of Fermat point to be pretty unhelpful and misleading. Specifically, you just want the usual "packing" property that no solution can be good (i.e. has low excess empirical loss) for two distinct databases $\mathcal{D}^{(i)}, \mathcal{D}^{(j)}$. Please consider just stating it in this form.
- You have stated in several places that the best known lower bound in the unconstrained approximate-DP case is $\Omega\left(\frac{\sqrt{p}}{\epsilon n \log p}\right)$ from [Asi et al., 2021]. I'm not sure why the lower bound of [Song et al., AISTATS 2021]  doesn't give $\Omega\left(\frac{\sqrt{p}}{\epsilon n}\right)$ here; can you please explain more? (In particular, can't you just set $rank = p$ in their paper?)
- Another high-level comment regarding unconstrained ERM: it is never made clear in the paper what the assumption is when you state e.g. that there is an algorithm that gives $O\left(\frac{\sqrt{p \log(1/\delta)}}{\epsilon n}\right)$. My understanding is that $\|\theta^* - \theta^{initial}\|_2$ can be at most $O(1)$ for this to hold. Please consider stating this more carefully (and also emphasize that such assumption holds for your lower bound instances).
- I would also recommend checking out the papers "Robust traceability from trace amounts" by Dwork et al. and "Tight Lower Bounds for Differentially Private Selection" by Steinke and Ullman. These two papers extend the fingerprinting code approach of Bun et al. and has results that might be closer to what you use in the approximate-DP lower bound. For example, Theorem 3 of Steinke-Ullman actually has a lower bound of the form of the correlation between the output vector and the bias of the mean; I'm suspecting that this might help simplify your approximate-DP proof a bit.
- Page 7 Lemma 3.1: Don't you need $\epsilon \leq 1$ here?
- Page 8: "Coordinate dictionary order" -> "lexicographic order"?
- Page 9 typo: "oversome" -> "overcome"

**Summary Of The Paper:**

## Summary of Contributions

This paper studies the unconstrained empirical risk minimization (ERM) under differential privacy (DP). In this setting, there is a loss function $\ell: \mathbb{R}^p \times \mathcal{X} \to \mathbb{R}$ and we are given $x_1, \dots, x_n \in \mathcal{X}$; the goal is to output $\theta$ that minimizes the empirical loss $L(\theta; X) := \frac{1}{n} \sum_{i=1}^n \ell(\theta; x_i)$. The goal is to minimize the excess empirical loss $\mathbb{E}[L(\theta; X) - \min_{\theta^* \in \mathbb{R}^p} L(\theta^*; X)]$. We want our algorithm to satisfies $(\epsilon, \delta)$-DP. Recall that the case $\delta > 0$ is referred to as *approximate-DP* whereas the case $\delta = 0$ is referred ti as *pure-DP*. Here we assume that $\ell$ is 1-Lipchitz; the results easily extends to $C$-Lipchitz functions with an extra multiplicative factor of $C$ in the excess empirical loss.

The main contributions of the paper are:
1. In the approximate-DP setting, the authors show a lower bound of $\Omega\left(\frac{\sqrt{p \log(1/\delta)}}{\epsilon n}\right)$. This improves upon the best known bound of $\Omega\left(\frac{\sqrt{p}}{\epsilon n \log p}\right)$ in the unconstrained case from [Asi et al., 2021] and $\Omega\left(\frac{\sqrt{p}}{\epsilon n}\right)$ in the constrained case [Bassily et al., FOCS 2014]. The new lower bound also matches the known upper bound in both cases [Bassily et al., FOCS 2014].
2. In the pure-DP setting, the authors show a lower bound of $\Omega\left(\frac{p}{\epsilon n}\right)$.

To prove 1., the authors reduce from the 1-way marginal problem (similar to previous work). Recall that in 1-way marginal, we are given $x_1, \dots, x_n \in \\{-1, 1\\}^p$ and the goal is to approximate $\frac{1}{n} \sum_{i=1}^n x_i$; a lower bound of $\Omega(\frac{\sqrt{p \log(1/\delta)}}{\epsilon n})$ is known for the problem [Bun et al., STOC 2014]. The authors use an $\\ell_1$-distance loss function, i.e., $\\ell(\\theta; x_i) = ||\\theta - x_i||\_1$. Notice that here the optimal solution is $\\theta^*\_j = sign(\sum\_{i} z\_{i, j})$. This in spirit is very similar to 1-way marginal but not exactly the same. Specifically, if $\sum_{i} z_{i, j}$ is roughly around zero, then taking $\theta_j = -1$ or $\theta_j = 1$ does *not* effect the loss too much. Therefore, a direct "blackbox" reduction from 1-way marginal does not seem to work. To overcome this, the authors observe that actually in the construction of [Bun et al., STOC 2014] most of the coordinates' means are not close to zero (formalized as "biased mean" property in the current paper) and thus the hard instance gives the desired lower bound for DP ERM.

To prove 2., the authors use a standard packing-style construction together with the $\\ell_2$-distance loss function

**Summary Of The Review:**

## Recommendation

Overall, although the techniques are somewhat similar to previous work, I still think that the problems considered in this paper are important enough that the contributions of the paper (i.e. closing the gaps in excess empirical loss) are above the bar for ICLR. However, due to the potential correctness issue with the pure-DP lower bound, I have to give a weak reject for now. (I'm opening to changing this score depending on whether this issue can be fix and how easy the fix is.)

---

> ### Author Response · Authors · 2021-11-15
> **Reply to Reviewer E1tY**
>
> Thanks for your careful review and helpful suggestions! We especially appreciate your pointing out a flaw in our pure-DP proof which we ignored.
>
> For the proof of the pure-DP lower bound,
> we did make a mistake and missed the second term in the calculation.
> After considering the second term $\min_{\theta} L(\theta;D)$, we can still get a dimension-dependent, but unfortunately untight bound $\Omega(\frac{p^2}{n^2 \epsilon^2})$.
> The crucial issue is that a private solution which is $\Omega(p/n\epsilon)$ far from the Fermat point can not lead to an $\Omega(p/n\epsilon)$ loss in some bad case under our construction.
>
> Our fix is quite simple. We abandon the $\ell_2$ loss and instead extend the linear loss used in [Bassily et al. 14] to $R^p$ which preserves the convexity and lipschitzness at the same time. The benefit of using this loss is that $\theta^*$ lies on the boundary of the unit ball and is easily tractable if we only add 0 as dummy points. Moreover, the average $q(D)$ of data points is equal to $\frac{n^*}{n} \theta^*$ so that we can directly make use of Lemma 5.1 in BST14 (change its construction by adding 0 as dummy points instead of $c,-c$ therein). Then we prove $L(\theta;D)-L(\theta^*;D)\ge \frac{n^*}{8n} ||\theta-\theta^*||_2^2$, and use a method similar to Theorem 5.2 in BST 14 finishes the proof.
> We upload the new proof as an independent supplement and would appreciate it if you could read it.
> Sorry for the mistake and for increasing your workload.
>
> For approximate-DP, we can't use the extension method as in the pure-DP case, because approximate-DP uses fingerprinting codes so that the dataset $D$ doesn't have the super regularity as in the pure-DP case where $D$ only contains $n^*$ identical vectors $d$ and $n-n^*$ $\mathbf{0}$ vectors. Then it's difficult or even impossible to
> lower bound the value $L(\theta;D)-L(\theta^*;D)$
> by $\|\theta-\theta^*\|_2$
> because this task is essentially similar to finding and discussing the Fermat point of many points without good structure in high dimension in general.
>
> As for your confusion on the lower bound of [Song et al., AISTATS 2021], consider the case when $p>>n$ please.
> As rank will be smaller than $\min\{n,p\}$, in this case the lower bound in [Song et al., AISTATS 2021] is $\Omega(\frac{1}{\epsilon\sqrt{n}})$, which is much smaller than the bound of Asi et al. and ours, say about $\Omega(\sqrt{p}/n\epsilon)$.
>
> We use the parameter $C$ to upper bound $|\theta^*-\theta^{initial}|_2$ in the statement, e.g. theorem 3.4, and our proof is considering the case when $C=\Theta(1)$ without loss of generality.
> We will make it more clear.
>
> We only use the case and present a proof when $\epsilon\leq 1$ for Lemma 3.1, though there is some similar conclusion for the case when $\epsilon>1$. We will make the statement more rigorously.
>
> We will take care of your other comments seriously, and would like to answer any other further questions you may have.

---

> > ### Comment · Reviewer_E1tY · 2021-11-19
> > **Re: Reply to Reviewer E1tY**
> >
> > I'd like to thank the authors for replying to my comments. They take care of my concerns, specifically the new supplementary material looks correct.
> >
> > One quick question: for the new loss (i.e. equation (6)), wouldn't it be true that if all points are in the unit ball, then the minimizer $\theta^*$ must also be in the unit ball? If so, then wouldn't the result then follow in a blackbox manner from Bassily et al. (because there is no point in outputting anything outside the unit ball and so we reduce back to the constrained case)?

---

> > > ### Author Response · Authors · 2021-11-20
> > > **Re: Re: Reply to Reviewer E1tY**
> > >
> > > Thanks for your reply.
> > >
> > > Your quick question is very good and we are also thinking about it recently.
> > > The minimizer should be in the unit ball, and if doing the projection will not increase the value of the function.
> > > We think the new method based on the Lipschitz extension can present a reduction from unconstrained case to constrained case, though there are some details that need attention and we haven't checked them carefully. If correct, this method can present an alternative and simple proof of the result obtained by our extension based on mean-biased property.

---

> > > ### Author Response · Authors · 2021-12-05
> > > **Extension to Non-Euclidean norm**
> > >
> > > We would like to emphasis one important advantage of our result, that it can be generalized to $\ell_q$ norm directly for any $q\geq 1$ for both the constrained and unconstrained cases.
> > > Thus it improved the lower bound $\Omega(\frac{\sqrt{p}}{n\epsilon \log p})$ for $\ell_1$ (Theorem 10. in [Asi et al. 2021]) and the lower bound $\Omega((q-1) \frac{\sqrt{p \log (1 / \delta)}}{\epsilon n})$ for the constrained $\ell_q$ where $1<q\leq 2$ (Theorem 7.2 in [Bassily et al. 2021]).
> > >
> > > Considering $\ell_q$ norm means that the objective functions of ERM are $G_q$ Lipschitz with respect to $\ell_q$, and the diameter $C_q$ of the domain is also defined with respect to $\ell_q$.
> > > Roughly speaking, the convex hull of the dataset we use in our original construction has an $\ell_{\infty}$-ball shape and the objective functions we used are $\ell(\theta;d)=\|\theta-d\|_1$. This nice property guarantees that the product of Lipschitz constant and diameter is always $G_qC_q=p^{1-\frac{1}{q}} p^{\frac{1}{q}}=p$ for any $q\geq 1$.
> > > We didn't find direct results for DP-ERM with respect to the $\ell_q$ norm when $q>2$, and the lower bound nearly matches the known upper bound (Theorem 12. in [Asi et al. 2021]) when $q$ varies in $[1,2)$.
> > > As for the construction of BST14, the convex hull of the dataset is roughly a weaker $\ell_2$ ball. The result of BST14 can generalize to $q<2$ but can not be generalized to $q>2$.
> > > In their construction $G_2=C_2=1$, but take $\ell_\infty$ as an example, directly applying their construction implies $G_\infty=\sqrt{p},C_\infty=1$, which will make the lower bound with respect to $\ell_\infty$ sub-optimal.
> > > Changing the $\ell_2$ unit ball used to $\ell_\infty$ unit ball, will incur other challenges and can not lead to optimal bounds with current techniques.
> > >
> > > In a word, our result presents lower bounds $\Omega(\frac{\sqrt{p\log(1/\delta)}}{\epsilon n})$ for all $q>=1$ and for both constrained case and unconstrianed case.
> > > Particularly, our bound is the best and the first for $q>2$ to our knowledge.

---

### Official Review · Reviewer_rseh · 2021-11-01

**Correctness:** 4
**Technical Novelty And Significance:** 2
**Empirical Novelty And Significance:** Not applicable
**Recommendation:** 3
**Confidence:** 4

**Main Review:**

Private ERM is a very basic problem in private machine learning, and it’s worth understanding completely the trade off between different parameters. The prior work of Bassily et al. gave an almost complete picture, and the results in this paper are meant to fill some gaps in that work. However, a number of the results are, unfortunately, not new, and have been published before. In particular, the authors seem to not be aware of the journal version of Steinke and Ullman’s paper, which appeared in the Journal of Privacy and Confidentiality (volume 7, no 2, 2016). Theorem 5.1 there gives the tight dependence on $\delta$ for constrained ERM with linear losses. Theorem 5.2 shows a similar lower bound for $\ell_2^2$ loss, and it’s trivial to see that the theorem holds for unconstrained ERM. Theorem 5.2 thus nearly matches one of the main results in this paper, and matches it exactly when the error $\alpha$ is required to be constant. The only shortcoming of Theorem 5.2 form Steinke-Ullman compared to the present paper is that the sample complexity is shown to depend on $\alpha$ as $1/\sqrt{\alpha}$, which is a result of them taking a strongly convex loss.

Another weakness is that, even where the results were not published before, the techniques are very similar to prior work.

**Summary Of The Paper:**

The paper describes some lower bounds for differentially private empirical risk minimization (DP ERM). The main results are:

* An $\Omega(\sqrt{p \log(1/\delta)}/\epsilon n)$ lower bound for unconstrtained ERM under approximate differential privacy. This improves on prior work of Bassily et al. in the presence of the $\log(1/\delta)$ term, and in the optimization being unconstrained.

* An $\Omega(p/\varepsilon n)$ lower bound for pure differential privacy.

The proofs use now standard techniques from the literature: fingerprinting codes for approximate DP and a packing argument for pure DP.

**Summary Of The Review:**

The paper proves some good-to-know lower bounds on private ERM, but a significant portion of the results have been published before. At the very least these prior results need to be discussed. I am not convinced there is enough novel material here to warrant publication.

---

> ### Author Response · Authors · 2021-11-15
> **Reply to Reviewer rseh**
>
> Thanks four your feedback!
> If we understand right, your main concern is the novelty of our work, and we will discuss the prior results in more details.
>
> We do miss the Journal version of Steinke and Ullman's paper and we appreciate your pointing this out.
> But we discussed this in the penultimate paragraph in our Sec 1.3, and admit that it is straightforward to combine Steinke and Ullman's technique and the construction in [Bassily et al. 14] for the tight bound in the constrained case.
> See the sentence ``Though with some effort...''.
> Besides, as Asi et al. 2021 shows, combining the group privacy and the results on one-way marginals can get a bound which is off a few logarithmic terms.
> We should make it more clear in our paper that the main challenges arise from getting the tight lower in the unconstrained case.
>
> Theorem 5.1 in Steinke and Ullman's Journal version still uses the linear functions, which is inappropriate in the unconstrained case as they can lead to negatively infinite values.
> We choose the $\ell_1$ loss as the new loss function, and directly
> utilizing group privacy on it will incur new problems, for which we propose the unbiased mean property.
> We kindly refer you to section 1.2.1 for detailed discussion on this.
>
> If you have any further question, or have some idea on simplifying the proof and get the tight bound directly from known results, we would be very happy to know that.
> We would appreciate it very much if you could consider a revaluation of our work.

---

> > ### Comment · Reviewer_rseh · 2021-11-15
> > **Theorem 5.2 in Steinke-Ullman**
> >
> > Thank you for the response. I agree that Theorem 5.1 in Steinke-Ullman only works for the constrained case. Theorem 5.2, however, which uses the squared $\ell_2$ loss, also works in the unconstrained case, even though it’s stated with a constraint. For constant $\alpha$, that theorem recovers the lower bound $\Omega(\varepsilon^{-1}\sqrt{d \log(1/\delta)})$. The dependence on $\alpha$ is not tight because they use a strongly concave function, so I see the main contribution of your result as getting a tight dependence on $\alpha$ in the unconstrained case.

---

> > > ### Author Response · Authors · 2021-11-15
> > > **Strongly convex case**
> > >
> > > Thanks for your quick reply.
> > >
> > > In fact, Theorem 5.2 in Steinke-Ullman is already tight for strongly convex functions in the constrained case.
> > > If we require the function to be both Lipschitz and strongly convex, as BST14 defines, it does not make sense to consider the unconstrained case.
> > > And as one can observe, both their Theorem 5.1 and Theorem 5.2 follow directly from the one-way marginals, so we do not think it can help the problem considered in our paper.

---

> ### Author Response · Authors · 2021-12-05
> **Extension to Non-Euclidean norm**
>
> We would like to emphasis one important advantage of our result, that it can be generalized to $\ell_q$ norm directly for any $q\geq 1$ for both the constrained and unconstrained cases.
> Thus it improved the lower bound $\Omega(\frac{\sqrt{p}}{n\epsilon \log p})$ for $\ell_1$ (Theorem 10. in [Asi et al. 2021]) and the lower bound $\Omega((q-1) \frac{\sqrt{p \log (1 / \delta)}}{\epsilon n})$ for the constrained $\ell_q$ where $1<q\leq 2$ (Theorem 7.2 in [Bassily et al. 2021]).
>
> Considering $\ell_q$ norm means that the objective functions of ERM are $G_q$ Lipschitz with respect to $\ell_q$, and the diameter $C_q$ of the domain is also defined with respect to $\ell_q$.
> Roughly speaking, the convex hull of the dataset we use in our original construction has an $\ell_{\infty}$-ball shape and the objective functions we used are $\ell(\theta;d)=\|\theta-d\|_1$. This nice property guarantees that the product of Lipschitz constant and diameter is always $G_qC_q=p^{1-\frac{1}{q}} p^{\frac{1}{q}}=p$ for any $q\geq 1$.
> We didn't find direct results for DP-ERM with respect to the $\ell_q$ norm when $q>2$, and the lower bound nearly matches the known upper bound (Theorem 12. in [Asi et al. 2021]) when $q$ varies in $[1,2)$.
> As for the construction of BST14, the convex hull of the dataset is roughly a weaker $\ell_2$ ball. The result of BST14 can generalize to $q<2$ but can not be generalized to $q>2$.
> In their construction $G_2=C_2=1$, but take $\ell_\infty$ as an example, directly applying their construction implies $G_\infty=\sqrt{p},C_\infty=1$, which will make the lower bound with respect to $\ell_\infty$ sub-optimal.
> Changing the $\ell_2$ unit ball used to $\ell_\infty$ unit ball, will incur other challenges and can not lead to optimal bounds with current techniques.
>
> In a word, our result presents lower bounds $\Omega(\frac{\sqrt{p\log(1/\delta)}}{\epsilon n})$ for all $q>=1$ and for both constrained case and unconstrianed case.
> Particularly, our bound is the best and the first for $q>2$ to our knowledge.

---

> > ### Comment · Reviewer_rseh · 2021-12-06
> > **Interesting, perhaps can be fleshed out more in a new version of the paper**
> >
> > Thank you for these additional details. It seems that the paper can be made significantly stronger than the initial submission if the content from the comments here is added to it and fleshed out. I am, however, generally reluctant to recommend acceptance based on such significant changes to the original submission.

---

### Official Review · Reviewer_FRcE · 2021-11-08

**Correctness:** 4
**Technical Novelty And Significance:** 3
**Empirical Novelty And Significance:** 3
**Recommendation:** 5
**Confidence:** 2

**Main Review:**

**Strengths:**

- The main strength of this paper is that it gives tight lower bounds for a topic that has been well studied in the literature. The bounds are presumably the last word on the subject.
- The lower bounds are proved via a use of fingerprinting codes. These have been used in the DP lower bound literature before. Here they introduce a new notion known as Error robust biased mean fingerprinting codes. As a non-expert in the area, it is unclear to me how much novelty exists in their definition and application here. But given the results are strong, I am not overtly concerned. I would suggest that the authors perhaps come up with a shorter name.

**Weaknesses:**
The central weakness is that the paper is very poorly written, starting from the choice of notation. For instance:
1. In the abstract, none of the quantities appearing in the lower bound have been defined.
2. One page 1, $\epsilon, \delta$ are not defined before the discussing of the gap in existing bounds in terms of $\delta$.
3. In discussing other paper's results in the introduction, terms like $\theta$ and $\alpha$ are used without any indication of what they stand for.
4. In the definition of Fingerprinting codes (Def 3.1), the quantity $d$ appears without any prior definition in specifying $F(C_S)$. A bit of digging around shows that this is really meant to be $p$.

Numerous typos like this together with the quality of exposition make this paper impossible for anyone who is not already and expert in the area, and can error-correct for the inaccuracies here.





**Summary Of The Paper:**

This paper presents tight lower bounds on differentially private ERM, a well-studied topic in the DP literature. It obtains tight bounds for both the constrained and unconstrained settings. The exact quantitative improvements are stated precisely in the abstract, and I won't repeat them here. The improvements are not staggering, but they are tight, and are presumably the final word on this topic.

**Summary Of The Review:**

It is a strong paper based on the results, but the current version is only going to make sense to experts in the area. It is not a bad paper to accept, but it seems like it would benefit from a substantial rewrite. I would label it borderline.

---

> ### Author Response · Authors · 2021-11-15
> **Reply to Reviewer FRcE**
>
> Thank you very much for your positive feedback and constructive criticism!
> Your feedback reminds us that we ignored the readability of this paper, and we apologize for the confusion in the notations and the typos which cause (potential) trouble to non-expert readers.
> We will rewrite the paper by defining the notations carefully, clarifying the typos and making the paper more friendly to readers not familiar with this topic in the future version.

---

### Decision · Program_Chairs · 2022-01-20

**Decision:**

Reject

**Comment:**

This paper gives sample complexity lower bounds for differentially private empirical risk minimization (ERM). While the reviewers agreed that the results are non-trivial, the general consensus was that the proofs are tweaks of previously developed techniques and that the main result is actually new in a rather narrow setting (specifically, for unconstrained ERM and sub-constant error parameter). Another concern was that one of the proofs (the one on pure differential privacy) was incorrect in the submission; a different proof was provided subsequently (which also closely follows prior work). Finally, the reviewers pointed out several issues with the clarity of the presentation and comparison to prior work. Given the above, this work is below the acceptance threshold.